# Are Pedobarographic and Gait Analyses Useful Tools to Evaluate Outcomes of Anterograde Calcaneo-Stop Procedure in Pediatric Symptomatic Flexible Flatfoot?

**DOI:** 10.3390/children9030366

**Published:** 2022-03-05

**Authors:** Daniela Dibello, Valentina Di Carlo, Federica Pederiva

**Affiliations:** 1Paediatric Ortopedics, Giovanni XXIII Children’s Hospital, 70126 Bari, Italy; 2Paediatric Orthopedics, Institute for Maternal and Child Health—IRCCS “Burlo Garofolo”, 34137 Trieste, Italy; valentinadicarlo3@gmail.com; 3Pediatric Surgery, Institute for Maternal and Child Health—IRCCS “Burlo Garofolo”, 34137 Trieste, Italy; federica_pederiva@yahoo.it

**Keywords:** gait analysis, symptomatic flexible flatfoot, pes planovalgus

## Abstract

Background: Flexible flatfoot is a frequent condition in childhood that needs to be treated when symptomatic. The aim of this study was to analyze pedobarographic and gait outcomes of patients with painful flexible flatfoot who underwent the anterograde calcaneo-stop procedure. Methods: All patients scheduled for surgical correction of painful flexible flatfoot between April and September 2011 were offered to participate in a study of dynamic pedobarographic and gait analyses before surgery and 3, 12, and 24 months after surgery. A healthy control group of similar age and physical characteristics also underwent dynamic pedobarographic and gait analyses. Results: Fifteen patients accepted to undergo dynamic pedobarography and gait analyses. The data were compared with fifteen controls of similar age and BMI. No significant differences were found on dynamic pedobarography within patients at different endpoints, except for a decreased percentage of plantigrade phase and increased percentage of digitigrade phase at 12 months post-op in comparison with 3 months post-op, nor when compared with control. Similarly, when range of motion was taken into consideration, no significant differences were found within patients at different endpoints, nor when compared with control, except for a decrease in ankle joint range of motion 24 months post-op in comparison with the controls. The stride was significantly decreased before surgery and became like controls 24 months after the procedure. The cadence, significantly decreased 24 months after surgery in comparison with the pre-surgical set, was similar to the controls. No significant differences were seen in the walking speed within patients at different endpoints and the controls. The cycle time significantly increased 24 months after surgery when compared to the pre-op situation, and was similar to the controls. Conclusion: Dynamic pedobarographic and gait analyses were useful not only to identify the gait impairment in patients with symptomatic flexible flatfoot, but also to measure the treatment outcome through the analysis of the surgery’s impact on the gait quality.

## 1. Background

Flatfoot is a progressive developmental or acquired deformity characterized by the flattening of the medial arch, plantar, and medial rotation of the talus and forefoot abduction. Flexible flatfoot is frequent in childhood and represents a normal medial arch during non-weight bearing and a flattening of the arch on stance. It has long been a controversial issue whether to treat or to observe flexible flatfoot. However, when the conservative approach fails and the children present with pain or fatigue after physical activity, there is general agreement that surgical correction must be considered [1,2]. Among the different types of surgical treatments, we prefer to perform the anterograde calcaneo-stop procedure [3] described by Castaman. As many other investigators [3,4,5], we believe that beyond an obvious mechanical effect, the calcaneo-stop procedure has a proprioceptive function.

Whereas many studies reported the clinical outcomes of flexible flatfoot surgical correction [1,4,6], the literature lacks in studies on the effect that these procedures have on the gait.

We aimed to analyze the dynamic pedobarographic and gait outcomes of our patients with painful flexible flatfoot who underwent the anterograde calcaneo-stop procedure.

## 2. Methods

After approval by the Institutional Review Board on 24 February 2011 (11/15), all patients who presented at our institution between April 2011 and September 2011 with painful flexible flatfoot and were scheduled for surgical correction were offered to participate in a study of dynamic pedobarographic and gait analyses to be performed before surgery and 3, 12, and 24 months after surgery. An informed consent was obtained from parents of the minors included in the study at the time in which the informed consent for the surgical correction was taken. Children with flatfoot caused by neurologic and muscular disorders, genetic conditions and syndromes, collagen disorders, ligamentous laxity, trauma, or with congenital flatfoot were excluded. 

All patients with symptomatic flexible flatfoot, defined by clinical and radiographic evaluation, and resistant to nonoperative treatment, underwent the anterograde calcaneo-stop procedure [3] performed under local anesthesia, without tourniquet, with or without sedation. The patient was placed supine, the operated extremity was rotated slightly inwards with a 90° bended knee and the supinated foot was kept by the assistant leaning on the fluoroscopic machine to allow clear fluoroscopic control. An incision of 1 cm was applied centered on the sinus tarsi. An entry hole was made into the talus with a trocar and a steel screw (VCA, Mikai^®^, Genoa, Italy) of the desired diameter and length (6.5 mm × 35 mm, 8 mm × 35 mm, or 8 mm × 40 mm) was percutaneously inserted at a 35° direction in the sagittal and 45° in the coronal plane (Figure 1). The dorsiflexion of the foot was checked with the knee in extended position. No cast immobilization was required. Patients were allowed for full weight bearing the same day of the surgery or as soon as possible. Foot exercises and active and passive ankle joint mobilization were recommended post-operatively. The screw was scheduled for removal 24–26 months after the first surgical procedure.

All patients agreeing to the study underwent dynamic pedobarographic and gait analyses at the aforementioned endpoints. A healthy control group of similar age and physical characteristics was considered and underwent dynamic pedobarographic and gait analyses as well.

The FootWork Pro pressure plate and its software (Amcube, Gargas, France) were used to collect pedobarographic data. The child was asked to walk on the mat at a self-selected speed. For each subject, a minimum of five passes per foot was obtained and the pressure loads on the feet were detected, observing how the feet perform the phase of the step. The percentage of time spent on each step phase (taligrade, plantigrade, and digitigrade) was recorded at four endpoints (before surgery and 3, 12, and 24 months after surgery) in the patients and compared with the control group. 

The range of motion of each joint (hip, knee, and ankle joint) was measured using a goniometer at four endpoints (before surgery and 3, 12, and 24 months after surgery) in the patients and compared with the control group.

For three-dimensional motion capture, eighteen passive reflective markers were attached according to the Lundberg skin markers protocol. The Pro Reflex Qualisys (Goteborg, Sweden) system, consisting of 6 infrared cameras, was used for the optical tracking of the markers. The gait was registered while the child walked along a walkway several times at a comfortable speed. The frequency of the motion capture was set at 120 frames per second (120 Hz). Qualisys Track Manager software (Goteborg, Sweden) was used for data processing. Four parameters were considered in patients before surgery as well as 3, 12, and 24 months after surgery and compared with control data: stride (m), cadence (steps/min), walking speed (cm/s), and cycle time (s).

The dynamic pedobarography, range of motion, and gait analyses data at four endpoints were compared within the cohort of patients and then with the control group, using a non-parametric Wilcoxon test.

Statistical analyses were conducted using the Software GraphPad Prism 6.0 (San Diego, CA, USA) and a *p* value < 0.05 was considered statistically significant.

## 3. Results

Forty-five patients underwent bilateral surgical correction of painful flexible flatfoot between April and September 2011 and were eligible for the study. All of them were third grade according to Viladot. Fifteen (8 male and 7 female) accepted to undergo dynamic pedobarography and gait analyses. They all underwent an anterograde calcaneo-stop procedure. All patients included in this study had the screw removed 24 months after the first surgical procedure. At the time of the final pedobarography and gait analyses, the patients had a median age of 14 years (range 11–16 years) and a median BMI of 19.5 (range 14–27). The same patients had a median age of 12 years (range 9–14 years) and a median BMI of 16.8 (range 15.3–17.4) at the beginning of the study. All surgical procedures were uneventful with no infections, screw displacement, or breakage. Neither gastrocnemius and soleus lengthening, nor calcaneal osteotomy was necessary.

A group of 15 controls of similar age (median 14 years, range 10–15 years) and BMI (median 21.5; range 15–23) was recruited.

The pedobarography data are shown in Table 1. The percentage of each step phase was first calculated in the control group and then compared with the patients at four endpoints. No significant differences were found within patients at different endpoints, except for a decrease of the percentage of the plantigrade phase and increase of the percentage of the digitigrade phase at 12 months post-op in comparison with 3 months post-op, with no significant differences occurring when compared with control. 

The range of motion data are displayed in Table 2. No significant differences were found within patients at different endpoints nor when compared with controls, except for a decrease in ankle joint range of motion at 24 months post-op in comparison with the controls.

Figure 2 shows the results of gait registrations. The stride was significantly decreased in the patients before surgery, and only 24 months after the procedure it became similar to the control. The cadence was significantly decreased 24 months after surgery in comparison with the pre-surgical set. However, no differences were found between the control group and patients after the screw was removed. No significant differences were seen in the walking speed between patients at different endpoints and the controls. The cycle time increased significantly after the screw was removed when compared to the pre-op situation, and was similar to controls. 

## 4. Discussion

Flexible flatfoot is the most prevalent condition seen in pediatric orthopedic clinics. The deformity is characterized by the absence of the medial arch, the protrusion of the head of the talus, and the valgus position of the calcaneus with weight bearing [4]. Symptomatic forms result in subjective complaints, deteriorate function, and produce substantial objective findings. The patients would complain about pain along the medial part of the foot, pain in the sinus tarsi, leg and knee, diminished endurance, gait disorders, prominent medial talar head, everted heels, and heel cord tightness. While an agreement on whether or not to treat asymptomatic flexible flatfoot has not been reached, all specialists agree that, in children presenting with pain or fatigue after physical activity, a surgical procedure must be considered [4]. Persistent pronation of the subtalar joint during the propulsive phase of gait is mostly responsible for major deformities in adult life [7].

The purpose of surgery is to restore and maintain physiological alignment between the talus and calcaneus, allowing the foot bones to remodel themselves during the subsequent period of growth. Arthroeresis limits subtalar joint pronation through the insertion of a screw into the sinus tarsi, which allows for the active correction by stimulating the proprioceptive foot receptors [6,8]. It is still controversial what the optimal age is at which to perform surgery. While some authors advocated that the best age for treatment is 12 years to avoid the development of cavovarus deformity in feet operated at a very early stage [9,10], others recommended surgery between 7–8 and 14–15 years [4]. Some surgeons argued that not enough correction could be reached in patients older than 14 years because of the limited bone growth potential [6].

Among different surgical techniques to correct symptomatic flexible flatfoot, we believe that the anterograde calcaneo-stop procedure [3] described by Castaman is a simple, reliable, and minimally invasive procedure, which allows for the alignment of the talus and calcaneus, restoring a proper foot arch. Moreover, as the correction is maintained even after the screw is removed, we join many other investigators [3,6,11] in claiming a proprioceptive function for this procedure, emphasizing proof that with this active self-correction, there is weak screw penetration in the talus and a relatively small percentage of screw breakage compared with what expected from a purely passive mechanical mechanism. The percutaneous procedure, most of the time, could be performed in our setting under simple local anesthesia and this made it even less invasive and reasonably straightforward. All our patients had satisfactory results without complications considering correction, pain, function, ability to return to physical activities, and patient’s personal satisfaction. All of them were allowed to walk immediately after surgery.

While the clinical results of the calcaneo-stop procedure, either in its traditional [4,7,11] or anterograde [1,6] form, have been widely published, the literature lacks in studies on the effect that these procedures have on the gait. Gait analysis provides a comprehensive amount of data that can be used to assess functional impairment and recommend interventions. It is a relevant and objective way to quantify the amount of impairment, and it correlates the movement of the hip, knee, and ankle joints. Moreover, we believe that it is a valuable tool to measure the success of the surgical procedure to correct symptomatic flexible flatfoot. 

The first step of our study was to select an age- and BMI-matched control group of healthy children with normal arched feet. The data from the control were then compared to patients to measure the degree of kinetic loss during gait due to symptomatic flexible flatfoot and the success of the surgical treatment in allowing normal values to be restored.

While comparing the same patients before and after surgery, no significant differences were found in the percentage of time spent on the taligrade step phase, but we observed a trend toward a decrement of the percentage as soon as the time of the screw removal was reached. In a different way, the percentage of the plantigrade phase of the step tended to increase after surgery, but then, at the time of the screw removal, the values were similar to the controls. Similarly, the digitigrade phase tended to increase after surgery, and 24 months post-op was similar to controls.

The range of motion of hip and knee joints after the removal of the screw was similar to controls. On the other hand, it was still significantly decreased in comparison with the control when the ankle joint was concerned. This might be due to the persistence of the functional block mechanically induced by the screw over 24 months. The functional adaptation of the ankle joint to the new setting seemed to happen more gradually and more time was required for the ankle to reach a normal range of motion. This calls for another assessment much later.

The analysis of gait before surgery confirmed that stride was significantly decreased in patients and increased gradually 12 months after surgery to regain normal values 24 months post-op. Similarly, the cycle time, significantly impaired in symptomatic flexible flatfoot, increased after surgery to reach a normal value. On the other hand, while the walking speed did not seem to be affected by the pathological condition, the cadence was significantly increased, and decreased after surgery till values were similar to controls. A patient with symptomatic flexible flatfoot walks with a shorter stride, increased cadence, slightly decreased speed, and decreased cycle time of the step. We previously demonstrated [12] that symptomatic flexible flatfoot is responsible for difficulties in walking and running, soreness and aching, as well as foot and ankle pain before surgical correction. Together with a subjective improvement of the overall situation after surgery, the gait analysis helped in demonstrating that the patients were objectively able to walk with a longer stride, decreased cadence, and longer cycle time. The walking speed seemed slower to catch up and this was probably related to the still decreased range of motion of the ankle joint. 

## 5. Conclusions

In conclusion, we believe that gait analysis is a reliable tool not only to provide the clinician with information to recommend surgery, but also to measure the treatment outcome and the surgical impact on the quality of the gait. We think that while it is probably too soon to have the gait analysis 3 months after the surgical correction, as the functional impairment is almost unchanged, it might be a good idea to perform it 12 months after surgery and 24 months post-op, after the removal of the screw. Our experience taught us that maybe a third endpoint 36 months after surgery should be considered to have a confirmation of the outcome.

## Figures and Tables

**Figure 1 children-09-00366-f001:**
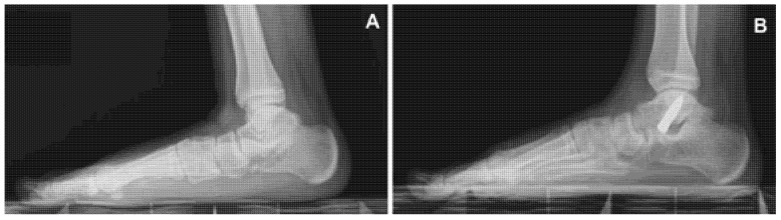
Symptomatic flexible flatfoot (**A**) treated with an anterograde calcaneo-stop procedure (**B**).

**Figure 2 children-09-00366-f002:**
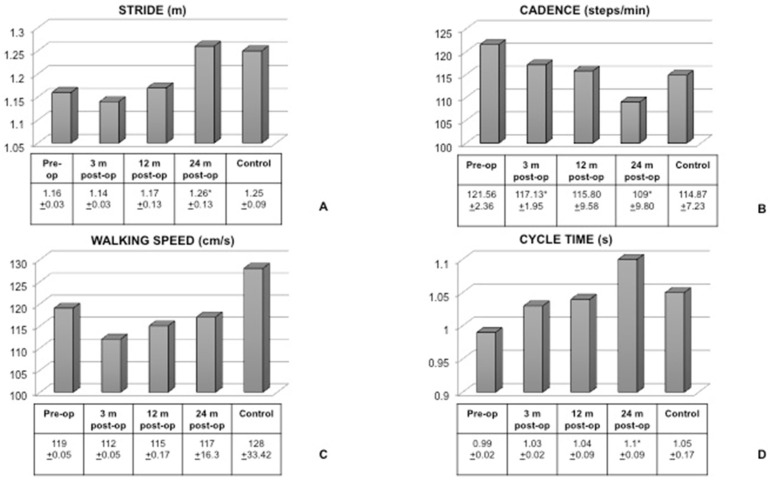
Gait analysis data. Stride (**A**), cadence (**B**), walking speed (**C**), and cycle time (**D**) were considered in patients before surgery and 3, 12, and 24 months after surgery and compared with the control. (**A**): * indicates *p* < 0.05 for 24 months post-op vs. pre-op. (**B**): * indicates *p* < 0.05 for 3 and 24 months post-op vs. pre-op. (**D**): * indicates *p* < 0.05 for 24 months post-op vs. pre-op. m = months.

**Table 1 children-09-00366-t001:** Pedobarography data. The percentage of each step phase for patients at four endopoints and for controls are displayed. * indicates *p* < 0.05 for 12 months post-op vs. 3 months post-op.

Step Phases	Pre-op (Mean ± SD)	3 Months Post-op (Mean ± SD)	12 Months Post-op (Mean ± SD)	24 Months Post-op (Mean ± SD)	Control (Mean ± SD)
Taligrade	23.70% ± 6.16	20.50% ± 8.80	19.32% ± 5.73	19.50% ± 6.84	19.63% ± 5.67
Plantigrade	37.58% ± 5.52	43.90% ± 13.70	39.80% * ± 11.15	40.22% ± 8.27	38.50% ± 6.67
Digitigrade	41.84% ± 5.74	38.70% ± 12.90	43.82% * ± 10.56	43.23% ± 8.21	45.60% ± 4.09

**Table 2 children-09-00366-t002:** The range of motion of the hip, knee, and ankle joints was measured in patients at four endpoints and compared with the control. * indicates *p* < 0.05 for 24 months post-op vs. control.

Range of Motion	Pre-op (Mean ± SD)	3 Months Post-op (Mean ± SD)	12 Months Post-op (Mean ± SD)	24 Months Post-op (Mean ± SD)	Control (Mean ± SD)
Hip joint	48.3° ± 6.2	43.8° ± 4.8	37.8° ± 16.0	44.0° ± 4.3	48.0° ± 5.8
Knee joint	70.9° ± 9.8	62.2° ± 3.1	55.33° ± 22.6	63.8° ± 3.7	63.2° ± 5.9
Ankle joint	32.6° ± 4.7	28.3° ± 3.9	26.6° ± 11.9	31.3° ± 3.9 *	35.4° ± 3.9

## Data Availability

The available data are reported on the manuscript.

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
