# Peer review of "Are Pedobarographic and Gait Analyses Useful Tools to Evaluate Outcomes of Anterograde Calcaneo-Stop Procedure in Pediatric Symptomatic Flexible Flatfoot?"

_children, 2022, doi:10.3390/children9030366_

Round 1

Reviewer 1 Report

It is an interesting and clinically relevant topic. This paper could be a good and important contribution to the evaluation of calcaneo-stop procedure in paediatric flexible flatfoot. The scientific quality of the manuscript is not sufficient in this form, and it needs thorough revision.

My comments can be found in the PDF document.

Reviewer 2 Report

Very interesting results.

Author Response

Thanks for your review

Round 2

Reviewer 1 Report

Thank you for the changes of your manuscript and the detailed response to the comments. It is an interesting and clinically relevant topic. This paper is a good and important contribution to the evaluation of calcaneo-stop procedure in paediatric flexible flatfoot.